crystallography/materials science/biomaterials

carbonated hydroxyapatite, microspheres, controllable size, EDTA

**Authors for correspondence:**
Mei-li Qi
e-mail: beauty0507@163.com
Yanmin Wang
e-mail: ymwangsdu@139.com

# Investigation of EDTA concentration on the size of carbonated flowerlike hydroxyapatite microspheres

Shengkun Yao[1], Mei-li Qi[2], Liang Qi[3], Yongling Ding[4], Min Chen[4] and Yanmin Wang[4]

[1]Shandong Provincial Engineering and Technical Center of Light Manipulations and Shandong Provincial Key Laboratory of Optics and Photonic Device, School of Physics and Electronics, Shandong Normal University, Ji'nan 250358, People's Republic of China
[2]Shandong Branden Medical Devices Co. Ltd, Qihe 251100, People's Republic of China
[3]Chaoyue Science and Technology Co. Ltd, Ji'nan 250100, People's Republic of China
[4]School of Transportation and Civil Engineering, Shandong Jiaotong University, Ji'nan 250357, People's Republic of China

M-liQ, 0000-0001-8857-7522; YW, 0000-0001-7149-507X

Ethylenediamine tetraacetic acid (EDTA) is considered an effective crystal growth modifier for template-assisted hydrothermal synthesis of hydroxyapatite (HA) materials. In this work, flowerlike-carbonated HA (CHA) microspheres were synthesized using EDTA via a one-step hydrothermal route. The phase, functional groups, morphology and particle size distribution of the products were examined by X-ray diffraction, Fourier transform infrared spectrometer, field emission scanning electron microscopy as well as laser diffraction particle size analysis. Results show that the morphology of the products can be well controlled by adjusting the EDTA concentration. With an increase of the EDTA concentration, the particle size of flowerlike microspheres decreased from tens of microns down to a few microns. The underlying mechanism for the morphological transition of CHA microspheres with different concentrations of EDTA under hydrothermal conditions is proposed. This work provides a simple way to controllably fabricate CHA microspheres with various sizes using the same synthesis system for biomedical applications, such as cell carriers and drug delivery.

## 1. Introduction

Carbonated hydroxyapatite (CHA) inherits excellent properties from hydroxyapatite (HA) and possesses superior solubility and biological performance [1]. According to the mode of carbonate

ion substitution, synthetic CHA can be divided into A-type (the hydroxyl position), B-type (the phosphate ion position) and AB-type (both the hydroxyl and phosphate ion position) [2,3]. Among various morphologies of CHA, porous CHA with a three-dimensional structure has aroused extensive attention in the tissue engineering field as bone graft substitutes and injectable bone repair materials owing to excellent biocompatibility, bioactivity and osteoconductivity [4,5]. Owing to the large surface area and excellent surface activity, flowerlike CHA microspheres show more potential applications, for example, drug loading/releasing systems and ion adsorption/exchange agents [6,7]. Moreover, the facile and low-cost synthesis of CHA microspheres allows them to be applicable for various biomedical applications.

Recently, extensive studies have been concentrating on enhancing the drug delivery and release characteristics of CHA microsphere-based systems through granularity control. Many methods have been reported to synthesize flowerlike CHA microspheres, mostly including the hard template casting route [8–11] and soft template self-assembly route [12,13]. As to the former method, multi-step and complicated processes need to be carried out and thus one template is usually not enough to adjust the pore size. The porous structure tends to be destroyed easily accompanied by the subsequent sintering process. These shortcomings make the hard template casting method an inappropriate candidate for porous CHA microspheres. However, the latter one which takes only one step is the proper one to prepare porous CHA microspheres. Obviously, it is more economical and more manipulable, compared with the hard template casting method. Although a variety of reagents, including poly(styrene sulfonate) [14], sodium citrate [15], cetyltrimethylammonium bromide [16], as well as ethylenediamine tetraacetic acid (EDTA) [17] have been used to adjust the final microstructure of CHA, the synthesis of porous CHA microspheres with controllable spherical degree using the same synthesis system remains a challenge. The size of microspheres determines their loading capacity and target location. Thus, it is demanding to develop CHA microspheres with controllable morphology based on their special usage.

In this study, EDTA with different concentrations was employed in a one-step hydrothermal preparation method of flowerlike CHA microspheres with different particle sizes. Urea, rather than $NH_3 \cdot H_2O$ in common use, was adopted as both a pH regulator and a $CO_3^{2-}$ source to prepare CHA microspheres. The influence of EDTA concentration on the sizes of the CHA microspheres is investigated and a possible mechanism of the formation of CHA microspheres with various particle sizes is put forward. This work provides an efficient way to realize the controllable sizes of CHA microspheres instead of changing the synthetic system.

## 2. Experimental

Flowerlike CHA microspheres with different sizes were obtained via hydrothermal synthesis. During the experiment, $Ca(NO_3)_2$ (0.01 mol l$^{-1}$), $(NH_4)_2HPO_4$ (0.06 mol l$^{-1}$) and urea (0.1 mol l$^{-1}$) were mixed homogeneously on a magnetic stirrer (Zhengzhou, Henan, China). Then EDTA (0 M, 0.01 M, 0.2 M, 2 M) was added into the above clear solution, respectively. After mixing well, dilute $HNO_3$ solution (volume ratio of nitric acid to deionized (DI) water = 1 : 2) were added to the mixture dropwise until the pH value reached 3.50. Finally, 80 ml of the reactants were moved to 100 ml stainless steel autoclaves and treated at 180°C for 5 h. After the reaction, the resultants were washed with DI water and ethanol thoroughly and dried at 80°C for 5 h.

With the help of an X-ray diffractometer (XRD; Bruker D8 Advance, Cu K$\alpha$ radiation source, $\lambda = 1.5418$ Å) and Fourier transform infrared (FTIR; Bruker Tensor 27) spectroscopy, the phase identification information and functional groups of the products were obtained. A field emission scanning electron microscope (FE-SEM; SU-70) was applied to characterize the products' morphology. Owing to the nonconductivity of the samples, they need to be sputtered with gold before the FE-SEM test. The particle size distribution (PSD) was evaluated by a laser particle analyser (LS13320) and DI water was chosen as the dispersion medium.

## 3. Results and discussion

### 3.1. Phase identification

XRD patterns of the products synthesized by using different concentrations of EDTA, together with the bar chart of standard HA (JCPDS no. 09–0432) are displayed in figure 1. All of the products have similar

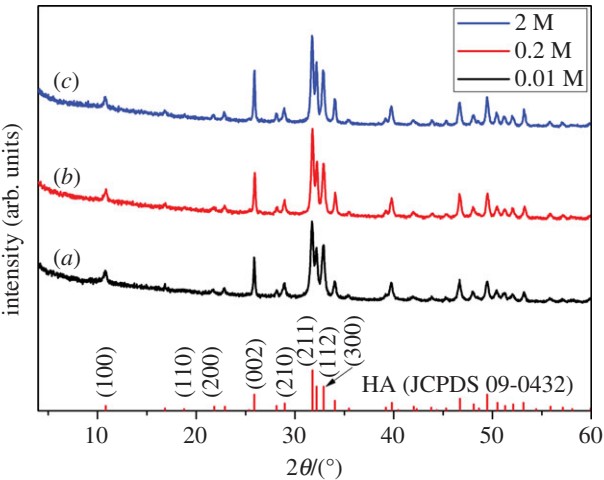

**Figure 1.** XRD patterns of the products with different concentrations of EDTA. (*a*) 0.01 M, (*b*) 0.2 M and (*c*) 2 M.

**Table 1.** Unit cell parameters of all the samples confirmed by XRD data.

| samples | unit cell parameters (nm) | | | c/a ratio |
|---|---|---|---|---|
| | a | b | c | |
| standard HA (PDF 09-0432) | 9.421 | 9.421 | 6.884 | 0.731 |
| HA regulated by 0.01 M EDTA | 9.436 | 9.436 | 6.886 | 0.730 |
| HA regulated by 0.2 M EDTA | 9.427 | 9.427 | 6.874 | 0.730 |
| HA regulated by 2 M EDTA | 9.437 | 9.437 | 6.879 | 0.729 |

XRD patterns, which can be indexed into the HA phase. No other phases are detected, suggesting that single-phase HA crystals can be hydrothermally synthesized by using 0.01–2 M EDTA as the template. The high and narrow diffraction peaks reflect the good crystallinity of HA. Changing the concentration of EDTA does not affect the crystalline phase of the products. The method is applicable to synthesize HA crystals using different concentrations of EDTA.

By calculating the unit cell parameters carefully, we found a slight change. The lattice parameters of all the samples are listed in table 1. Compared with stoichiometric HA, the replacement of the large tetrahedral $PO_4^{3-}$ ion by the small planar $CO_3^{2-}$ ion leads to a variation of the unit cell parameters in the crystal: decrease in a-axis and increase in c-axis. The c/a ratio will increase owing to the formation of B-type CHA, while A-type substitution shows the opposite [3]. However, the c/a ratios in this study exhibit a slight decrease. This could be attributed to the replacement of $PO_4^{3-}$ as well as $OH^-$ by $CO_3^{2-}$ simultaneously.

## 3.2. Functional group analysis

To examine the functional groups, FTIR tests were performed on all the samples and the corresponding results are shown in figure 2. Figure 2*a–c* shows the FTIR spectra of the as-obtained samples with 0.01 M, 0.2 M and 2 M EDTA, respectively. It can be seen that all the samples have similar spectra. Peaks at around 1115, 1032 and 960 cm$^{-1}$ are attributed to asymmetric and symmetric stretching vibrations in the $PO_4^{3-}$ group [18]. Peaks at 602 and 563 cm$^{-1}$ are produced by the bending modes of the O-P-O bonds of $PO_4^{3-}$ group [19]. The absorption peaks at about 3572 and 632 cm$^{-1}$ which were not obviously found belong to the stretching and flexural modes of $OH^-$ group, respectively, further demonstrating the A-type $CO_3^{2-}$ replacement on $OH^-$ sites [20]. Peaks at 1630 cm$^{-1}$ and the broad absorption peak at 3420 cm$^{-1}$ arise from the flexural and stretching vibration of the adsorbed water [21]. In addition, peaks at 1456 and 874 cm$^{-1}$ are produced by the vibration of the $CO_3^{2-}$ group,

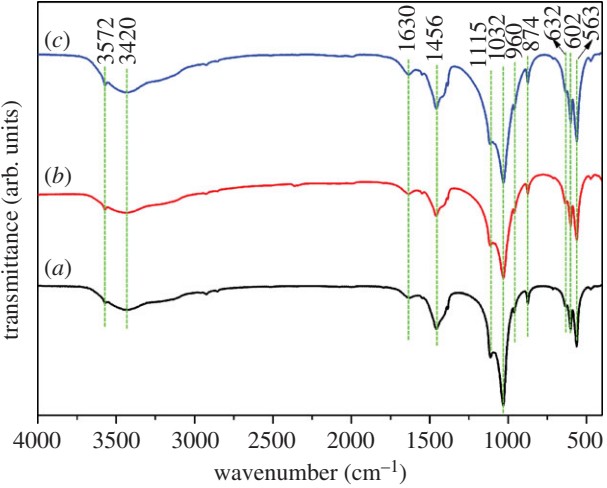

**Figure 2.** FTIR spectra of the products with different concentrations of EDTA. (*a*) 0.01 M, (*b*) 0.2 M and (*c*) 2 M.

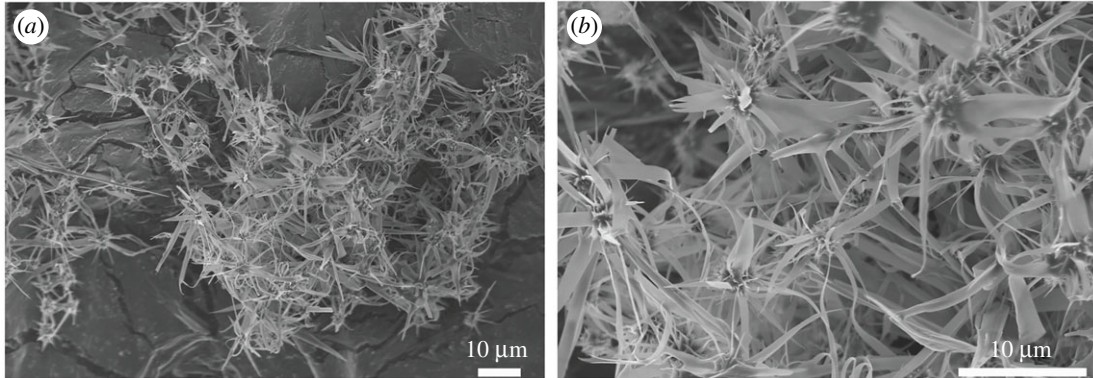

**Figure 3.** FE-SEM images of the products without adding EDTA at different magnifications. (*a*) 1000 × and (*b*) 3000 ×.

which may be derived from the dissolved carbon dioxide in the reaction solution, suggesting that $CO_3^{2-}$ replaces the B-site of $PO_4^{3-}$ in HA crystal [22,23]. Because $CO_3^{2-}$ ions are generated during the hydrolysis process of urea in the hydrothermal synthesis, the incorporation of a small amount of $CO_3^{2-}$ into HA crystals is inevitable [24]. Based on the above results, all the as-obtained products are confirmed to be AB-type carbonated HA (CHA) phase.

## 3.3. Microstructural characterization

Figure 3 shows the FE-SEM images of the sample without adding EDTA. It is indicated that the products contain three-dimensional structures formed by irregular fibre arrays.

Morphologies of the samples with different concentrations of EDTA at different magnification times are demonstrated in figure 4. From the low magnifications of the CHA samples prepared by EDTA at 0.01 M, 0.2 M and 2 M in figure 4*a*, *c* and *e*, respectively, we can see that the products are mainly made up of flowerlike microspheres, indicating the key role of EDTA in regulating the morphology of the samples (compared with that in figure 3). As the concentration of EDTA increases, the roundness of the microspheres become better and the size of the microspheres become smaller. The average granularity of the microspheres prepared by EDTA at 0.01 M and 0.2 M is around 30 μm and 25 μm, while that of the microspheres prepared by EDTA at 2 M significantly decreases to about 3 μm. Also, the regular assemblies of nanoflakes into porous microspheres are observed, apparently indicating that the EDTA template plays a dominant role in inducing CHA into growing directionally, and the concentration of EDTA is a key parameter to regulate the size of the CHA microspheres. EDTA can chelate free $Ca^{2+}$ in the reaction system, mostly the Ca channels along the c-axis. Thus, a higher concentration of the template can dissolve c-planes quicker and restrict the growth more effectively, leading to the directional growth of CHA and finally the formation of smaller CHA microspheres.

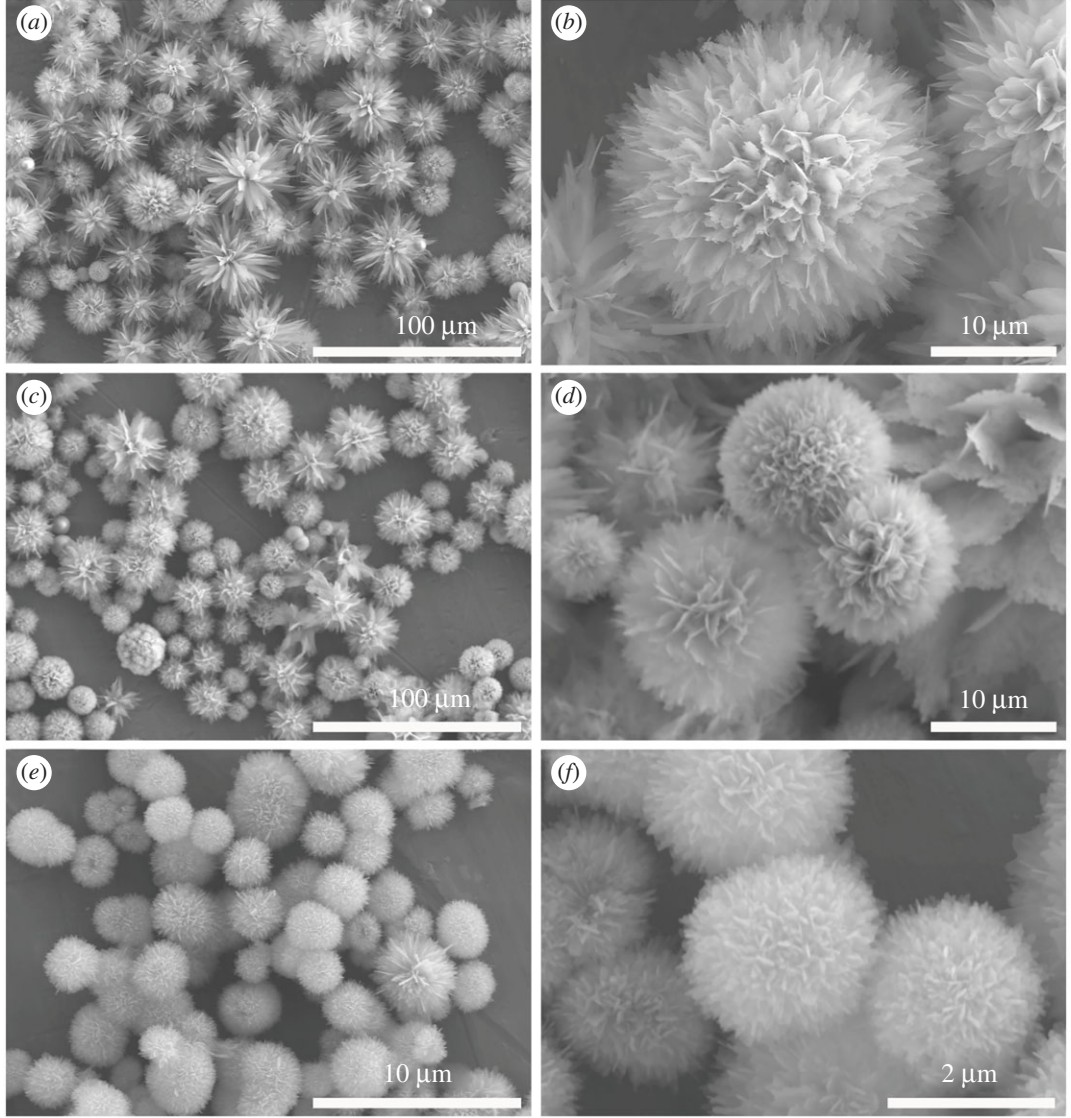

**Figure 4.** FE-SEM images of carbonated hydroxyapatite (CHA) products with different concentrations of EDTA. (*a,b*) 0.01 M, (*c,d*) 0.2 M and (*e,f*) 2 M.

## 3.4. Particle size distribution

PSD largely influences the properties of CHA microspheres. The PSD results of the three samples can be found in figure 5. The appearance of the main size peaks represent that the size of the CHA microspheres is relatively uniform. It is evident that the peaks shift to the left with the increase of concentration of EDTA, indicating the particle size of the microspheres has a tendency to decrease. Compared with the other two curves, the particle size of the sample with 2 M EDTA mainly centres at 1–10 µm (the blue curve in figure 4). The transition of such changes confirms that the amount of EDTA added is crucial to the construction of CHA materials. Statistical analysis results of all the samples are summed up in table 2. In the table, d10, d50 and d90 represent 10%, 50% and 90% of the total particles which are smaller in diameter compared with the listed values, respectively. It is concluded that variation of the mean particle size is in accordance with that in figure 5. The PSD changes among different samples by using different concentrations of EDTA and are in line with the FE-SEM results.

## 3.5. Formation mechanism

Based on previous analysis, a possible formation mechanism accounting for the morphological transformation of flowerlike CHA microspheres with different concentrations of EDTA in the

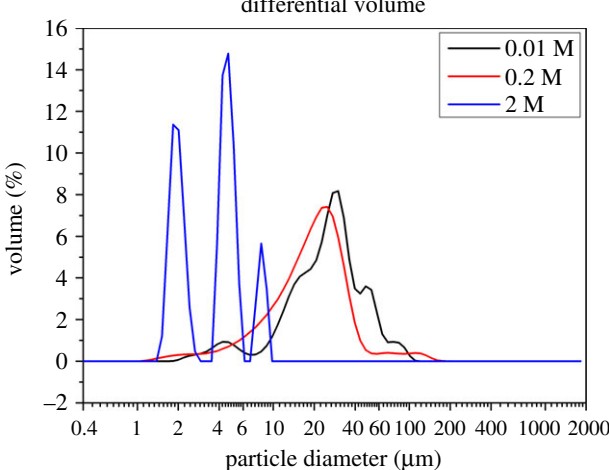

**Figure 5.** Particle size distribution of the CHA microspheres with different concentrations of EDTA.

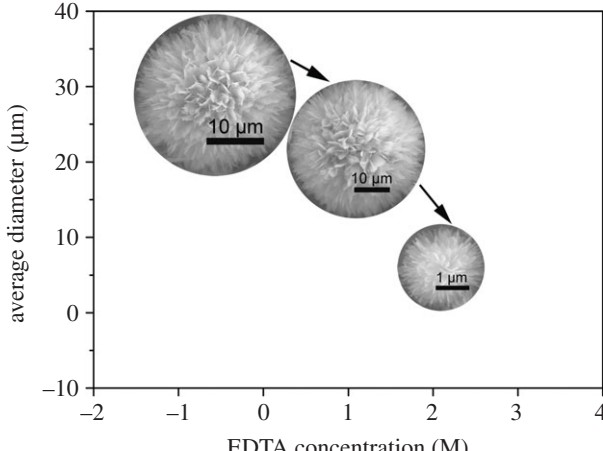

**Figure 6.** Relationship between the diameter of the flowerlike CHA microspheres and the EDTA concentration.

**Table 2.** Particle sizes of flowerlike CHA microspheres.

| the concentration of EDTA | d10 (μm)[a] | d50 (μm)[b] | d90 (μm)[c] | mean (μm) |
|---|---|---|---|---|
| 0.01 M | 11.00 | 27.14 | 54.21 | 29.93 |
| 0.2 M | 7.60 | 20.73 | 37.35 | 23.74 |
| 2 M | 1.872 | 4.379 | 7.923 | 4.18 |

[a]10% of the total particle size which is less than some value.
[b]50% of the total particle size l which is less than some value.
[c]90% of the total particle size which is less than some value.

hydrothermal process is proposed. Figure 6 demonstrates the relationship between the diameter of the flowerlike CHA microspheres and the test conditions. The formation of flowerlike CHA microspheres with various sizes is the dual role of urea's hydrolysis action and EDTA's inhibiting effect of the active growth parts onto CHA's surface.

First, the EDTA will chelate with $Ca^{2+}$ ions in the solution when the hydrothermal reaction has not yet begun. The stability of the Ca–EDTA complex improves with the increased pH and decreases with the increased reaction temperature [25]. The chelated $Ca^{2+}$ is originally unstable because of the low pH value (initial pH = 3.5), leading to the explosive release of free $Ca^{2+}$. The initial pH value defines the

morphology of the products by influencing the nucleation rate as well as the growth kinetics of the final crystal [26]. With the hydrolysis action of urea (above 80°C), the solution pH begins to increase and $CO_3^{2-}$ occurs. In this case, negative $HPO_4^{2-}$, $OH^-$ and $CO_3^{2-}$ ions will combine with the free $Ca^{2+}$ ions released from the Ca–EDTA complex and produce CHA nuclei. At the initial stage, masses of free $Ca^{2+}$ lead to the relatively high supersaturation in the solution, which generates a large amount of CHA nuclei. These nuclei tend to self-assemble with each other and thus adopt a flowerlike structure to reduce the total surface energy [27], being similar to the self-assembly and organization of cystine microcrystals into superstructures and nonclassical crystallization [28,29]. The EDTA template can chelate the $Ca^{2+}$ ions assigned largely in Ca channels along the c-axis of CHA crystal. The preferential growth of the c-axis is restricted effectively, which causes CHA into growing directionally in the final stage. The higher concentration of the EDTA template, the quicker they dissolve c-planes. Thus, smaller CHA microspheres are finally obtained.

## 4. Conclusion

Carbonated flowerlike HA microspheres with different sizes are synthesized by a one-step hydrothermal route with different concentrations of EDTA chelating agent. The phase of the products doesn't change under the control of EDTA. As the concentration of EDTA increases, the porous microspheric morphology assembled by flakes is maintained. However, the size of the microspheres decreased by degrees, especially at 2 M EDTA. The higher concentration of the EDTA template can dissolve c-planes of CHA crystals more quickly, leading to the restriction of preferential growth along the c-axis effectively and finally the formation of smaller CHA microspheres. A feasible formation mechanism of carbonated flowerlike HA microspheres for the morphological transformation adjusted by different concentrations of EDTA under hydrothermal conditions is proposed. The route provides an easy way to realize the size controllable synthesis of flowerlike CHA microspheres without changing the reaction system and expand the bio-applications of HA materials such as the delivery of drugs, proteins and genes.

Data accessibility. All the data files of XRD patterns, FTIR spectra and PSD curves reported in this article can be found here (https://doi.org/10.5061/dryad.69p8cz911) [30].

Authors' contributions. S.Y. conceived of the study, designed the study and drafted the manuscript; M.L.Q. conducted the analysis and critically revised the manuscript; L.Q. coordinated the study and critically revised the manuscript; Y.D. helped with the tests; M.C. helped to carry out the experiments and draft the manuscript; Y.W. reviewed and critically revised the manuscript. All authors gave final approval for publication.

Competing interests. We declare we have no competing interests.
Funding. This work is financially supported by the National Natural Science Foundation of China (11947117, 12004227 and 51803109), the Natural Science Foundation of Shandong Province (ZR2020QE070 and ZR2020QA076) and China Postdoctoral Science Foundation (2019M660164).

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
