## [Peer Review File · Royal Society Open Science]

Review History

RSOS-202148.R0 (Original submission)

Review form: Reviewer 1

Is the manuscript scientifically sound in its present form?

No

Are the interpretations and conclusions justified by the results?

Yes

Is the language acceptable?

Yes

Do you have any ethical concerns with this paper?

No

Have you any concerns about statistical analyses in this paper?

No

Recommendation?

Major revision is needed (please make suggestions in comments)

Comments to the Author(s)

Reviewer Report

Manuscript ID: RSOS-202148

Journal: Royal Society Open Science

Title: Investigation of EDTA concentration on the size of carbonated flowerlike hydroxyapatite microspheres

This work discusses the role of EDTA-crystal growth modifier on carbonated hydroxyapatite to produce flower like microsphere in one-step synthesise route. These microspheres of hydroxyapatite get attracted as it can be used for drug delivery application. Producing uniform microspheres are challenging however the authors have successfully identified and used the crystal growth modifier to perform this synthesis. Therefore, this work may be considered for publication in Royal Society Open Science if the authors are agreed to revise the manuscript on below comments and suggestions.

1. In the experimental section, it could useful to add drying time at 80 °C.
2. It could be better to use 'nonconductivity' and 'sputtered' instead of inconductivity and sprayed.
3. In the FTIR spectra, is any changes in the stretching and bending vibration of PO₄³⁻ due to B-site occupation of carbonate ions?. A reference should be given for the sentence 'the incorporation of a small amount of CO₃²⁻ into HA crystals is inevitable',
4. Either FTIR or SEM, the pure CHA spectra/image should be compared with EDTA-CHA. Importantly, this will give clear evidence for the influence of EDTA on CHA nonetheless dual role of urea and EDTA.
5. The mechanism proposed here may follow non-classical crystallization as your experimental condition creates high supersaturation and produce hierarchical structure. " These nuclei tend to self-assemble with each other and thus adopt a flowerlike structure to reduce the total surface energy". The authors may refer below papers for nonclassical crystallization and flower-like structure. (1) <https://doi.org/10.1021/jacs.9b01883>, J. Am. Chem. Soc. 2019, 141, 26, 10120-10136 (2) <https://doi.org/10.1039/C3CE42634C>, CrystEngComm, 2014,16, 4183-4193, (3) [dx.doi.org/10.1021/cg300935k](https://doi.org/10.1021/cg300935k), Cryst. Growth Des. 2012, 12, 4995-5001

Review form: Reviewer 2**Is the manuscript scientifically sound in its present form?**

Yes

Are the interpretations and conclusions justified by the results?

Yes

Is the language acceptable?

Yes

Do you have any ethical concerns with this paper?

No

Have you any concerns about statistical analyses in this paper?

No

Recommendation?

Accept with minor revision (please list in comments)

Comments to the Author(s)

This manuscript reports the synthesis of flowerlike carbonated hydroxyapatite microspheres in the presence of different concentrations of EDTA via a one-step hydrothermal route. The reported data are in general reliable, and the manuscript can be accepted for publication after revision.

1. Please give the pH values in the mixture solution after adding 0.01, 0.2 or 2 M EDTA.
2. Will the replacement of phosphate with carbonate affect the XRD spectrum of the samples in comparison with the standard XRD spectrum of pristine hydroxyapatite?
3. Comments on the format and details.
Figure 1. The label on the x-axis should be modified to be displayed completely.
Figure 2. There should be blank space after wavenumber.
Figure 5. The x-axis needs to be modified.

Decision letter (RSOS-202148.R0)

Dear Dr Qi:

Title: Investigation of EDTA concentration on the size of carbonated flowerlike hydroxyapatite microspheres
Manuscript ID: RSOS-202148

The editor assigned to your manuscript has now received comments from reviewers. We would like you to revise your paper in accordance with the referee and Subject Editor suggestions which can be found below (not including confidential reports to the Editor). Please note this decision does not guarantee eventual acceptance.

Please submit your revised paper before 31-Jan-2021. Please note that the revision deadline will expire at 00.00am on this date. If we do not hear from you within this time then it will be assumed that the paper has been withdrawn. In exceptional circumstances, extensions may be possible if agreed with the Editorial Office in advance. We do not allow multiple rounds of revision so we urge you to make every effort to fully address all of the comments at this stage. If deemed necessary by the Editors, your manuscript will be sent back to one or more of the original reviewers for assessment. If the original reviewers are not available we may invite new reviewers.

On behalf of the Subject Editor Professor Anthony Stace and the Associate Editor Dr Dattatray Late.

RSC Associate Editor:
Comments to the Author:
Major Revision needed.

RSC Subject Editor:
Comments to the Author:
(There are no comments.)

Reviewers' Comments to Author:
Reviewer: 1

Comments to the Author(s)
Reviewer Report
Manuscript ID: RSOS-202148

Journal: Royal Society Open Science
Title: Investigation of EDTA concentration on the size of carbonated flowerlike hydroxyapatite microspheres

This work discusses the role of EDTA-crystal growth modifier on carbonated hydroxyapatite to produce flower like microsphere in one-step synthesise route. These microspheres of hydroxyapatite get attracted as it can be used for drug delivery application. Producing uniform microspheres are challenging however the authors have successfully identified and used the crystal growth modifier to perform this synthesis. Therefore, this work may be considered for publication in Royal Society Open Science if the authors are agreed to revise the manuscript on below comments and suggestions.

1. In the experimental section, it could useful to add drying time at 80 °C.

2. It could be better to use 'nonconductivity' and 'sputtered' instead of 'inconductivity' and 'sprayed'.
3. In the FTIR spectra, is any changes in the stretching and bending vibration of PO₄³⁻ due to B-site occupation of carbonate ions?. A reference should be given for the sentence 'the incorporation of a small amount of CO₃²⁻ into HA crystals is inevitable',
4. Either FTIR or SEM, the pure CHA spectra/image should be compared with EDTA-CHA. Importantly, this will give clear evidence for the influence of EDTA on CHA nonetheless dual role of urea and EDTA.
5. The mechanism proposed here may follow non-classical crystallization as your experimental condition creates high supersaturation and produce hierarchical structure. " These nuclei tend to self-assemble with each other and thus adopt a flowerlike structure to reduce the total surface energy". The authors may refer below papers for nonclassical crystallization and flower-like structure. (1) <https://doi.org/10.1021/jacs.9b01883>, J. Am. Chem. Soc. 2019, 141, 26, 10120–10136 (2) <https://doi.org/10.1039/C3CE42634C>, CrystEngComm, 2014,16, 4183-4193, (3) [dx.doi.org/10.1021/cg300935k](https://doi.org/10.1021/cg300935k), Cryst. Growth Des. 2012, 12, 4995-5001

Reviewer: 2

Comments to the Author(s)

This manuscript reports the synthesis of flowerlike carbonated hydroxyapatite microspheres in the presence of different concentrations of EDTA via a one-step hydrothermal route. The reported data are in general reliable, and the manuscript can be accepted for publication after revision.

1. Please give the pH values in the mixture solution after adding 0.01, 0.2 or 2 M EDTA.
2. Will the replacement of phosphate with carbonate affect the XRD spectrum of the samples in comparison with the standard XRD spectrum of pristine hydroxyapatite?
3. Comments on the format and details.
Figure 1. The label on the x-axis should be modified to be displayed completely.
Figure 2. There should be blank space after wavenumber.
Figure 5. The x-axis needs to be modified.

Author's Response to Decision Letter for (RSOS-202148.R0)

See Appendix A.

Decision letter (RSOS-202148.R1)

Dear Dr Qi:

Title: Investigation of EDTA concentration on the size of carbonated flowerlike hydroxyapatite microspheres

Manuscript ID: RSOS-202148.R1

It is a pleasure to accept your manuscript in its current form for publication in Royal Society Open Science. The chemistry content of Royal Society Open Science is published in collaboration with the Royal Society of Chemistry.

On behalf of the Subject Editor Professor Anthony Stace and the Associate Editor Dr Dattatray Late.

RSC Associate Editor
Comments to the Author:
(There are no comments.)

Reviewer(s)' Comments to Author:

Appendix A

Dear Editors and Reviewers,

We would like to thank the editors for immediately processing and two reviewers for careful reviewing and making constructive suggestions concerning our manuscript entitled “*Investigation of EDTA concentration on the size of carbonated flowerlike hydroxyapatite microspheres*” (ID: RSOS-202148). All comments are valuable and very helpful for revision and improvement of our manuscript, and provide guidance to our future research. We have studied the comments carefully and have fully addressed all points raised by the reviewers in our revised manuscript. Please see the red text in our edited manuscript. Below is a point-by-point response to the reviewers’ comments and suggestions.

Response to the Reviewer 1:

Comment 1: *In the experimental section, it could useful to add drying time at 80 °C.*

Response: Thank reviewer 1 for the good suggestion. We should be more explicit while writing the manuscript. In the revised manuscript, we added the drying time of 5 h (please see line 75, page 4).

Comment 2: *It could be better to use ‘nonconductivity’ and ‘sputtered’ instead of inductivity and sprayed.*

Response: Thank reviewer 1 for pointing this out. We strongly agree with reviewer 1 that the words “nonconductivity” and “sputtered” are more appropriate than inductivity” and “sprayed” that we used. Considering reviewer 1’s suggestion, we have made corrections in the revised manuscript (please see line 80-81 on page 4).

Comments 3: *In the FTIR spectra, is any changes in the stretching and bending vibration of PO_4^{3-} due to B-site occupation of carbonate ions? A reference should be given for the sentence ‘the incorporation of a small amount of CO_3^{2-} into HA crystals is inevitable’*

Response: Thank reviewer 1 for the comment. By referring literature and analyzing the FTIR results carefully, we found that there is no evident change in the stretching and bending vibration of PO_4^{3-} ions due to B-site occupation of CO_3^{2-} ions [1-5]. Carbonate substitution consequentially induces crystal lattice aberrance, reflected by the XRD spectrum. The replacement of the large tetrahedral PO_4^{3-} ions by the small planar CO_3^{2-} ions leads to a variation of the unit cell parameters in the crystal: decrease a-axis and increase c-axis. The c/a ratio will increase due to the formation of B-type CHA, while A-type substitution shows the opposite. In the revised manuscript, we have

fully addressed the changes caused by CO_3^{2-} ions in “3.1. Phase Identification” part (line 93-104, Page 4-5).

Thank reviewer 1 so much for providing significant guidance, which could be a good direction for our future study.

According to reviewer 1’s suggestion, we have given a reference to support the sentence “the incorporation of a small amount of CO_3^{2-} into HA crystals is inevitable” [6].

[1] Lafon J P, Champion E, Bernache-Assollant D. Processing of AB-type carbonated hydroxyapatite $\text{Ca}_{10-x}(\text{PO}_4)_{6-x}(\text{CO}_3)_x(\text{OH})_{2-x-2y}(\text{CO}_3)_y$ ceramics with controlled composition[J]. Journal of the European Ceramic Society, 2008, 28(1): 139-147.

[2] Sun R, Yang L, Zhang Y, et al. Novel synthesis of AB-type carbonated hydroxyapatite hierarchical microstructures with sustained drug delivery properties[J]. CrystEngComm, 2016, 18(41): 8030-8037.

[3] Lin K, Zhou Y, Zhou Y, et al. Biomimetic hydroxyapatite porous microspheres with co-substituted essential trace elements: surfactant-free hydrothermal synthesis, enhanced degradation and drug release[J]. Journal of Materials Chemistry, 2011, 21(41): 16558-16565.

[4] He Q, Huang Z, Liu Y, et al. Template-directed one-step synthesis of flowerlike porous carbonated hydroxyapatite spheres[J]. Materials Letters, 2007, 61(1): 141-143.

[5] Lin K, Liu P, Wei L, et al. Strontium substituted hydroxyapatite porous microspheres: surfactant-free hydrothermal synthesis, enhanced biological response and sustained drug release[J]. Chemical engineering journal, 2013, 222: 49-59.

[6] Jevtic M.; Mitric M.; Skapin S.; Jancar B.; Ignjatovic N.; Uskokovic D. Crystal structure of hydroxyapatite nanorods synthesized by sonochemical homogeneous precipitation. Cryst. Growth Des., 2008, 8: 2217-2222.

Comments 4: *Either FTIR or SEM, the pure CHA spectra/image should be compared with EDTA-CHA. Importantly, this will give clear evidence for the influence of EDTA on CHA nonetheless dual role of urea and EDTA.*

Response: We would like to thank reviewer 2 for providing significant guidance. We strongly agree with reviewer 1 that the original morphology of HA before EDTA addition need to be shown. Considering reviewer 1’s suggestion, the blank control group was added in our revised manuscript. FE-SEM tests were conducted (Fig. 3, page 7) and corresponding explanation (line 125-129 and 133-134, page 7) was also included. FE-SEM images displays that the product contains a 3D structure formed by irregular fiber arrays. The influence of EDTA on the morphology of the products is evident. Thank reviewer 1 for guiding us to make our manuscript stronger and more convincing.

Comments 5: *The mechanism proposed here may follow non-classical crystallization as your experimental condition creates high supersaturation and produce hierarchical structure. “These nuclei tend to self-assemble with each other and thus adopt a flowerlike structure to reduce the total surface energy”. The authors may refer below papers for nonclassical crystallization and flower-like structure. (1) <https://doi.org/10.1021/jacs.9b01883>, J. Am. Chem. Soc. 2019, 141, 26, 10120–10136*

(2) <https://doi.org/10.1039/C3CE42634C>, *CrystEngComm*, 2014,16, 4183-4193, (3) [dx.doi.org/10.1021/cg300935k](https://doi.org/10.1021/cg300935k), *Cryst. Growth Des.* 2012, 12, 4995–5001

Response: We would like to thank reviewer 1 for providing these papers. These papers are closely related to the mechanism of our work. We read them carefully and cited them in the revised manuscript (line 187-190, page 10 and references 27-29).

Special thanks to reviewer 1 for the good comments.

Response to Reviewer 2:

Comment 1: *Please give the pH values in the mixture solution after adding 0.01, 0.2 or 2 M EDTA.*

Response: Thanks to reviewer 2 for pointing this out. The initial pH is very important in the preparation of HA. In our experiment, the solution pH was adjusted to 3.5 by dilute nitric acid after adding different concentrations of EDTA. EDTA was first added to chelate with Ca^{2+} in the solution and then the solution pH was adjusted to a same value. In the revised manuscript, we clarified the order of reagent addition (line 70-72, page 3).

Comment 2: *Will the replacement of phosphate with carbonate affect the XRD spectrum of the samples in comparison with the standard XRD spectrum of pristine hydroxyapatite?*

Response: Thank reviewer 2 for raising this point. The replacement of PO_4^{3-} with CO_3^{2-} will affect the XRD spectrum of the samples in comparison with the standard XRD spectrum of HA. Synthetic CHA has been classified as A-type (CO_3^{2-} substitutes OH^-) and B-type (CO_3^{2-} substitutes PO_4^{3-}) depending on the mode of carbonate substitution^[1]. Compared to stoichiometric HA, the replacement of the large tetrahedral PO_4^{3-} ion by the small planar CO_3^{2-} ion leads to a variation of the unit cell parameters in the crystal: decrease a-axis and increase c-axis. The c/a ratio will increase due to the formation of B-type CHA, while A-type substitution shows the opposite^[2].

Although no major difference exists in our XRD spectrum, there is still a slight change observed in the unit cell parameters. The unit cell parameters of the products prepared by different concentrations of EDTA confirmed by XRD data are listed in Fig. 1 and Tab. 1. However, the c/a ratio of the products prepared in this study shows a slight decrease. This could be attributed to the replacement of PO_4^{3-} together with OH^- by CO_3^{2-} simultaneously.

In the FTIR results, the absorption peaks at about 3572 and 632 cm^{-1} which were not obviously found belong to the stretching and flexural modes of OH^- group, respectively, further demonstrating the A-type CO_3^{2-} replacement on OH^- sites.

Based on the above results, all the as-obtained products are confirmed to be AB-type carbonated HA (CHA) phase. In the revised manuscript, we clarified this point, added a table and modified the description of “B-type CHA” to “AB type CHA” (Line 93-99, 112-114 and 121, Table 1, Page 4-5). Thanks to reviewer 2 so much for guiding us to explore more and finally give the right conclusion.

Figure 1. The XRD results of full spectrum fitting in all of the samples.

Table 1. Unit cell parameters of all the samples.

Samples	Unit cell parameters (nm)			c/a ratio
	a	b	c	
Standard HA (PDF 09-0432)	9.421	9.421	6.884	0.731
HA regulated by 0.01 M EDTA	9.436	9.436	6.886	0.730
HA regulated by 0.2 M EDTA	9.427	9.427	6.874	0.730
HA regulated by 2 M EDTA	9.437	9.437	6.879	0.729

[1]. Jiang Y, Yuan Z, Huang J, et al. Substituted hydroxyapatite: a recent development. *Materials Technology*, 2019: 1-12.

[2]. Landi E, Celotti G, Logroscino G, et al. Carbonated hydroxyapatite as bone substitute. *Journal of The European Ceramic Society*, 2003, 23(15): 2931-2937.

Comments 3: *Comments on the format and details.*

Figure 1. The label on the x-axis should be modified to be displayed completely.

Figure 2. There should be blank space after wavenumber.

Figure 5. The x-axis needs to be modified.

Response: Thank reviewer 2 for these good comments. We should be more careful to the format and details. According to reviewer 2's suggestions, we have modified these figures. The label on the x-axis in Figure. 1 has been displayed completely, blank space has been added after wavenumber in Figure. 2, and the x-axis of the last figure has been modified in the revised manuscript.

Special thanks to reviewer 2 for the good comments.

We tried our best to improve the manuscript and made some changes in the manuscript. These changes will not influence the content and framework of the paper. Here, we did not list the changes but marked the changes in red in the revised paper. We earnestly appreciate the editors' work, and hope that we have addressed the reviewers' concerns properly so that the manuscript can be accepted for publication in *Royal Society Open Science*.

Finally, we would like to thank the editor and reviewers once again for the useful suggestions, which help us a lot to improve the quality of our manuscript.